# Bimetallic AgPt Nanoalloys as an Electrocatalyst for Ethanol Oxidation Reaction: Synthesis, Structural Analysis, and Electro-Catalytic Activity

**DOI:** 10.3390/nano13081396

**Published:** 2023-04-18

**Authors:** Fabian Mares-Briones, América Higareda, Jose Luis Lopez-Miranda, Rubén Mendoza-Cruz, Rodrigo Esparza

**Affiliations:** 1Centro de Física Aplicada y Tecnología Avanzada, Universidad Nacional Autónoma de México, Boulevard Juriquilla 3001, Santiago de Querétaro 76230, Qro., Mexico; fabianmares@gmail.com (F.M.-B.); lopezfim@gmail.com (J.L.L.-M.); 2Unidad de Energía Renovable, Centro de Investigación Científica de Yucatán A.C., Carretera Sierra Papacal-Chuburná Puerto, Km 5, Sierra Papacal, Mérida 97302, Yuc., Mexico; ame_libert.16@hotmail.com; 3Instituto de Investigaciones en Materiales, Universidad Nacional Autónoma de México, Circuito Exterior S/N, Circuito de la Investigación Científica, C.U., Ciudad de México 04510, CDMX, Mexico; rmendoza@materiales.unam.mx

**Keywords:** AgPt nanoparticles, nanoalloy, structural characterization, electrocatalysis, ethanol oxidation reaction

## Abstract

In the present work, the chemical synthesis of AgPt nanoalloys is reported by the polyol method using polyvinylpyrrolidone (PVP) as a surfactant and a heterogeneous nucleation approach. Nanoparticles with different atomic compositions of the Ag and Pt elements (1:1 and 1:3) were synthesized by adjusting the molar ratios of the precursors. The physicochemical and microstructural characterization was initially performed using the UV-Vis technique to determine the presence of nanoparticles in suspension. Then, the morphology, size, and atomic structure were determined using XRD, SEM, and HAADF-STEM techniques, confirming the formation of a well-defined crystalline structure and homogeneous nanoalloy with an average particle size of less than 10 nm. Finally, the cyclic voltammetry technique evaluated the electrochemical activity of bimetallic AgPt nanoparticles supported on Vulcan XC-72 carbon for the ethanol oxidation reaction in an alkaline medium. Chronoamperometry and accelerated electrochemical degradation tests were performed to determine their stability and long-term durability. The synthesized AgPt (1:3)/C electrocatalyst presented significative catalytic activity and superior durability due to the introduction of Ag that weakens the chemisorption of the carbonaceous species. Thus, it could be an attractive candidate for cost-effective ethanol oxidation compared to commercial Pt/C.

## 1. Introduction

The use of alcohols as fuels in electrochemical cells has increased over the last decade due to their potential application in self-sustainable energy production with minimal environmental impact [1,2,3,4]. In addition, liquid fuels are safer and more convenient for storage and transportation than compressed hydrogen. Furthermore, for their high efficiency, fast refueling, and low outgassing, direct ethanol fuel cells (DEFCs) are considered future portable power conversion devices [5,6,7,8,9]. Platinum (Pt) is one of the most important noble metals for manufacturing electrocatalysts used in DEFCs [10]. However, its commercialization is limited by its high cost and decreased oxidation potential due to the ethanol crossover effect (ECO), electrocatalyst poisoning by chemisorbed intermediate species during transport, and oxidation of ethanol through the membrane, which results in the obtaining mixed potentials for the ethanol oxidation (EOR) and oxygen reduction (ORR) reactions, with a significant loss of their performance [11,12,13,14].

In recent years, particular attention has been paid to designing cost-effective electrocatalysts with excellent resistance to carbonaceous intermediate species poisoning without compromising the high catalytic activity of EOR [15,16]. Furthermore, implementing heterogeneous Pt-based nanostructures in nanoalloys form with controlled size and morphology has improved suppression in the adsorption of poisonous species to monometallic Pt [17]. Due to the above, several noble metals (Rh, Pd, Au, Ni, Ru), as oxophilic components with relatively stable and considerable reactivity, have been used to form Pt nanoalloys, facilitating the formation of oxygen species at lower potentials to effectively oxidize intermediate species through the Langmuir–Hinshelwood mechanism and improve electrocatalyst stability [18,19,20,21].

Moreover, silver (Ag) has been proposed as a functional and beneficial element with high chemical stability. It has an electronic structure and an atomic radius similar to that of Pt, fulfilling the conditions for forming substitutional solid solution rules; furthermore, their implementation in forming bimetallic nanostructures can modify the electronic valence bands, minimizing activation energies and developing new interfaces promoting electronic exchange between elements.

The AgPt alloy during the ethanol oxidation reaction can simultaneously promote the oxidation of intermediates through the activation of the surface sites of Pt to reduce its poisoning by CO-like species due to weak surface adsorption originating from introducing Ag atoms to the system [22,23,24,25]. Therefore, the modification in the stoichiometric composition will affect the local elemental arrangement, affecting the electronic structure of the superficial sites of Pt and modifying its stability and catalytic activity. In addition, control of the size and morphology during the synthesis of the alloy can result in an increase in the surface area of the material and a preferential atomic arrangement, improving, in the same way, its catalytic performance [26].

Based on the considerations described above, this study presents the development of AgPt electrocatalysts using an easy and flexible synthesis method based on the polyol technique to obtain alloyed nanoparticles with a controlled size and morphology. In addition, by varying the molar ratio of the precursors, the relationship between the composition and its catalytic behavior toward the ethanol oxidation reaction was studied. Electrochemical measurements confirmed that AgPt nanoparticles with different molar ratios exhibit an enhanced electrocatalytic performance compared to commercial Pt/C, having the highest performance, with AgPt (1:3)/C showing better catalytic activity and stability.

## 2. Materials and Methods

### 2.1. Materials

AgPt bimetallic nanoalloys, as well as pure Ag-seeds, were synthesized using silver nitrate (AgNO_3_, 99.99%) and potassium tetrachloroplatinate (II) (K_2_PtCl_4_, 99.99%) as metal precursors. All nanoparticles (NPs) were synthesized using ethylene glycol (EG 99.8%), as a reducer agent and stabilized with poly(N-vinylpyrrolidone) (PVP, M = 40,000 g mol^−1^). Reagent-grade acetone and isopropyl alcohol were used to carry out nanoparticle washing. For the electrochemical tests, solutions of potassium hydroxide (KOH, ≥85%) in deionized water (Milli-Q, 18 MΩ·cm) were used as an alkaline medium, and reagent-grade ethanol (CH_3_CH_2_OH, ≥99.5%). For the electrochemical inks, a solution of Nafion^®^ 117 perfluorinated resin (5% in water), and isopropyl alcohol ((CH_3_)_2_CHOH, 99.5%). Commercial platinum on Vulcan XC-72 carbon (20 wt% of filler, PRD.0.ZQ5.10000029756, Appendix A) was used as a reference sample. All chemicals were purchased from Sigma-Aldrich (Sigma-Aldrich, St. Louis, MO, USA).

### 2.2. Synthesis Procedures

The AgPt bimetallic nanoalloys were prepared using the seed-mediated growth polyol method. Ag seeds were first synthesized by adding 2 mL of AgNO_3_ (20 mM) and 4 mL of PVP (50 mM) into ten aliquots every 2.5 min to 10 mL of EG previously heated at 160 °C under magnetic stirring and maintaining the reaction for 1 h [27]. Subsequently, to synthesize the nanoalloys AgPt (1:1), 0.5 mL of Ag seeds were dispersed in 5 mL of EG at 160 °C. Then, 1 mL of AgNO_3_ (50 mM) and 2 mL of PVP (50 mM) were added to the dispersed Ag seed solution in 10 aliquots every 2.5 min under magnetic stirring. Finally, 1 mL of metal precursor K_2_PtCl (50 mM) and 2 mL of PVP (50 mM) were added to the solution using a procedure similar to that mentioned above by aliquots, ending the reaction with a residence time of 1h at 160 °C under magnetic stirring and a last heating ramp at 190 °C for 15 min. A similar procedure was used to synthesize the AgPt nanoalloys (1:3), modifying only the volumes of the precursor agents K_2_PtCl_4_ to 3 mL and twice the surfactant agent PVP (6 mL). The resulting nanoalloys were washed four times; the sample was diluted with acetone on every wash and centrifugated at 5000 rpm for 5 min. The supernatant was subsequently removed, and the sample was redispersed by ultrasound using isopropyl alcohol. Next, electrocatalysts were prepared by dispersing a known amount of AgPt NPs solution with 32 mg of Vulcan XC-72 carbon by ultrasonic (UP200Ht, 200 W, 26 kHz) for 5 min; the amount of the NPs was calculated to be 20 wt% of the total metal phase.

### 2.3. Structural Characterization

The post-synthesized AgPt NPs were investigated using a UV-Vis spectrophotometer Metash UV6000 (Shanghai Metash Instruments Co., Shanghai, China) in the 200–800 nm range. The structural characterization was performed with a Rigaku Ultima IV diffractometer (Rigaku Co., Tokyo, Japan) using the powder XRD technique in a 2θ range from 30° to 90° at room temperature. The Hitachi SU8230 cold-field emission scanning electron microscope CFE-SEM (Hitachi High-Tech Co., Tokyo, Japan) and Jeol JEM-ARM200F scanning transmission electron microscope TEM/STEM (JEOL Ltd., Tokyo, Japan) were used to analyze the structural, morphological, and chemical properties.

### 2.4. Electrochemical Characterization

The electrochemical measurements were performed on a BioLogic VSP potentiostat (Biologic, Seyssinet-Pariset, France) coupled with a standard electrochemical cell in a three-electrode configuration: a Hg/HgO electrode filled with NaOH (1.0 M) solution and a graphite bar, used as a reference and counter electrode, respectively, and a working electrode prepared by depositing 6 µL of catalytic ink on a glassy carbon BASi electrode (3 mm in diameter). The suspension consisted of 1 mg of the corresponding electrocatalyst, 70 µL of isopropyl alcohol, and 7 µL of Nafion^®^117 (Sigma-Aldrich, St. Louis, MO, USA) perfluorinated resin solution (5 wt%) dispersed by ultrasound for 15 min.

At least three different electrochemical profiles were carried out by cyclic voltammetry (CV) in a potential window of −0.80 to 0.85 V vs. NHE at a scan rate of 50 mVs^−1^ to inquire into the oxidation-reduction processes. The electrochemically active surface area (*ECSA*) was calculated using Equation (1), and the respective experimental errors were obtained.
(1)$$ECSA({cm}^{2}{mg}^{-1})=\frac{Q\left(mC\right)}{Q^{o}(mC\;{cm}^{-2})*M_{Pt\;or\;Ag}(mg)}$$
where *Q* is the electric charge corresponding to the integrated hydrogen adsorption/desorption region after subtracting the double-layer capacitive current for pure Pt/C, and AgPt/C bimetallic surface or the oxide reduction peak for pure Ag/C, *Q^o^* corresponds to the specific charge, 0.21 mC cm^−2^_Pt_ is used for one-electron transfer reaction of Pt and 0.40 mC cm^−2^_Ag_ [28] to convert the formation of one monolayer of AgOH/Ag_2_O to the surface area of Ag.

Electrochemical tests were performed to evaluate the ethanol oxidation reaction (EOR) using an aqueous solution of 1.0 M ethanol plus 0.3 M KOH as a supporting electrolyte, unless otherwise indicated. The catalytic performance was evaluated by CV at a scan rate of 20 mVs^−1^ and a potential window of −0.85 and 0.65 V vs. NHE for the AgPt/C system. In addition, the stability and CO-like tolerance were researched by chronoamperometry (CA) measurements for the ethanol oxidation process during 3600 s in the half-peak potential (Ep/2). Afterward, two potential cycles in a fresh electrolyte solution were recorded from the lower potential limit toward the anodic direction for the oxidative desorption of the intermediated species. In addition, cyclic stability toward the EOR to inquire about the long-term durability of the electrocatalysts was performed through an accelerated electrochemical degradation test that consisted of 2500 sweeps of cyclic potential in the same range of potential at a scan rate of 100 mVs^−1^.

Before each electrochemical measurement was performed, the adequate aqueous solution was deaerated by bubbling nitrogen-purging for at least 10 min. After that, all electrochemical tests were carried out at room temperature. The potential was reported with respect to a normal hydrogen electrode (NHE).

## 3. Results

The crystallographic nature of the synthesized bimetallic nanoparticles (BNPs) was characterized using the X-ray diffraction (XRD) technique. Typical XRD patterns of the AgPt alloy BNPs are shown in Figure 1. All XRD patterns showed three broad diffraction peaks that were assigned to (111), (200), and (220) reflections indexed with the face-centered cubic (fcc) crystal structure with an Fm-3m space group. No other diffraction peaks were observed in the second phase. These three peaks were observed at different 2θ compared to pure Ag (JCPDS 04-0783) and Pt (JCPDS 04-0802) structures. The XRD patterns displayed diffraction peak shifts linear to higher 2θ as the amount of Pt increased in the AgPt BNPs. This indicated that AgPt (1:3) had a higher composition of Pt than AgPt (1:1). Therefore, the results suggest that the AgPt BNPs are uniformly alloyed. The miscibility of an alloy was determined according to the Hume–Rothery rules [29], which indicate that the alloy is preferred when the crystal structure, atomic radii, valence, and electronegativity of the two elements are similar; therefore, Ag and Pt are capable of forming AgPt alloys. Furthermore, the corresponding phase diagram of the elements confirmed that the Ag-Pt system formed miscible heterogeneous alloys with atomic contents rich in Pt for the concentrations used in this study (1:1 and 1:3) [30]. Likewise, according to theoretical and experimental studies, the miscibility and diffusion improve with a decrease in particle size [31]. However, in some bimetallic nanosystems, it is also necessary to include the surface energy of the elements to determine their miscibility [32]. Vegard’s law can be an indication of miscible binary alloys that form solid solution (Equation (2)) [33]:(2)$${a=a}_{2}\left(1+\frac{a_{1}-a_{2}}{a_{2}}\times x_{1}\right)$$
where *a* is the calculated lattice parameter of AgPt BNPs, *a*_1_ and *a*_2_ are lattice parameters of two corresponding pure metals, Ag (0.4086 nm) and Pt (0.3923 nm), respectively, and *x*_1_ is the atomic fraction of component 1 (0.5 for AgPt (1:1), and 0.25 for AgPt (1:3)). The experimental lattice parameter was calculated from the Rietveld refinement, obtaining 0.3996 nm and 0.3965 nm for AgPt (1:1) and AgPt (1:3), respectively. According to Vegard’s law, the average calculated atomic composition was Ag_45_Pt_55_, and Ag_26_Pt_74_, these values were corroborated by energy dispersive X-ray spectroscopy (EDX) analysis. The average crystallite size was determined based on the broad peak of the (111) plane using Scherrer’s formula (Equation (3)) [34]:(3)$$t=\frac{0.9\lambda }{\beta cos\theta }$$
where *t* is the crystallite size, 0.9 is a shape factor that is an attribute of the equipment, *λ* is the X-ray wavelength (0.154 nm), *β* is the full width at half maximum intensity, and *θ* is the Bragg angle. The crystallite size values of the AgPt BNPs increased when the Pt content increased, from 6 nm to 10 nm for AgPt (1:1) and AgPt (1:3), respectively.

Figure 2a shows the results of the analysis using UV-Vis spectroscopy. Several metal nanoparticles showed the surface plasmon surface (SPR) phenomenon. For Ag, this signal was between 350 and 600 nm, the position of which depended on the size and morphology of the nanoparticles [35,36]. An intense absorption band located at 401 nm was observed in the spectrum corresponding to the Ag seeds. According to some reports, this position indicates the presence of small particles [37,38].

On the other hand, the spectrum corresponding to the AgPt (1:1) bimetallic nanoparticles showed a significant decrease in the signal corresponding to the Ag seeds. This decrease was more evident when the Pt concentration increased, as observed in the spectrum corresponding to the AgPt (1:3) sample. This behavior is because Pt does not show the SPR phenomenon in the UV-Vis spectrum. Therefore, the alteration of the shape and intensity of the band located at 407 nm depended on the combination of Pt with Ag seed leads. Sometimes, the Ag absorption band was shifted, wider, or disappeared completely [39]. Some very weak signals can be distinguished in the spectra shown in the figure. However, it was not possible to determine whether they corresponded to Ag. Therefore, the derivation process was used to corroborate the type of synthesized nanoparticles, specifically the fourth derivative with which absorption bands masked by other signals were distinguished. Figure 2b shows the results of the fourth derivative. As can be seen, in the graph corresponding to the Ag seeds, the same signal was observed in the UV-Vis spectrum at 401 nm. The graph of the AgPt (1:1) sample showed a signal at 406 nm due to the low concentration of Pt, without altering the position of Ag. Finally, the graph of AgPt (1:3) showed a signal at 407 nm, indicating the presence of Ag on the surface of the nanoparticles. The movement of the signal was attributed to the size increase by adding Ag and Pt atoms to the Ag seeds. All considerations mentioned above suggest the synthesis of bimetallic nanoalloys. Concerning the catalytic properties, the formation of AgPt nanoalloys in different atomic compositions can significantly modify the geometric factor and electron ligand effects, defining the number of atoms oriented in preferential directions and modifying the electron density distribution by the formation of missing links, thereby improving the selectivity and stability of the nanoalloy for specific catalytic reactions [40,41].

Aberration-corrected scanning transmission electron microscopy (STEM) was utilized for the detailed characterization of the AgPt BNPs. STEM is an invaluable tool for the characterization of nanostructures, mainly using the high-angle annular dark field (HAADF) detector, which is associated with the atomic number (Z) of the atoms on the specimen. The contrast in this imaging technique is strongly dependent on the chemical composition (Z-contrast). Therefore, HAADF-STEM allows easy identification of the elements present in the sample, elemental composition, and crystal information at an atomic scale. However, in this particular case, it was difficult to distinguish Ag and Pt as individual atomic columns by Z-contrast in the image because the atoms were randomly distributed, forming an alloy in contrast with the ordered alloys, where it was possible to distinguish individual atomic columns by Z-contrast in the images [42]. Figure 3a shows an HAADF-STEM image of the AgPt (1:3) BNP. From the image, d-spacings of 0.2299 and 0.1991 nm were obtained, which corresponded to (111) and (020) planes and were directly revealed in the atomic HAADF-STEM image. Meanwhile, the corresponding fast Fourier transform (FFT) reflects the fcc structure along the [101] zone axis (Figure 3b). It is clear that a well-defined truncated octahedron shape, enclosed by the {111} and {100} facets, appeared in the HAADF-STEM image. The geometry of the BNP was in accordance with the projection of a truncated octahedron model along the same direction (Figure 3c). The truncated octahedral shape is a highly symmetric structure enclosed by six {100} and eight {111} crystallographic surfaces. The surface energy density of {111} planes is the most stable, followed by {100} planes, and the other planes are relatively unstable [43]. Due to the catalytic reaction that occurs on the surface of the electrocatalyst, it is important to synthesize nanoparticles with the most stable planes [44]. A recent theoretical study reported that the exposed {111} crystal planes help C–C bond cleavage during ethanol oxidation, resulting in a higher fuel utilization [45].

The statistics-based atom-counting method has been applied to the atomic resolution image by using the StatSTEM program [46]. The statistics-based method relies on the total intensity of scattered electrons for each projected atomic column. The number of atoms in each projected atomic column is shown in Figure 3d, where the elements Ag and Pt distribution are presented randomly on its surface to form an alloy in a 1:3 relation with a total count of 6932 atoms, indicating the approximated atom composition for these size nanoparticles. The 2D thermal diagram corresponding to the thickness of the nanoparticle indicates that the entire perimeter has approximately 4 to 6 atoms; subsequently, a step of 10 and 12 atoms is generated until reaching the center of the particle with a thickness of 16 atoms according to the intensity bar. From these results, the physicochemical behavior of the nanoalloy for heterogeneous catalysis reactions can be discussed.

The energy dispersive X-ray spectroscopy (EDX)-STEM technique was applied to investigate the elemental distribution compositional line profiles on the AgPt (1:3) BNPs. The distributions of Ag and Pt are shown in Figure 3e, which reveals the homogeneous distribution of Ag and Pt elements across the entire BNP, showing the formation of the AgPt alloy structure.

The size and distribution of AgPt BNPs were determined by cold field emission scanning electron microscopy (SEM/STEM). Appendix A shows the low-magnification BF-STEM images obtained from the AgPt (1:1) and AgPt (1:3) BNPs. The average particle sizes were 6.3 nm and 10.4 nm, respectively. The increase in particle size was directly related to the increase in the volume of the precursor salts used during the synthesis procedure. Figure 4a,c shows the SEM images of AgPt (1:1) and AgPt (1:3) BNPs supported on Vulcan XC-72 carbon, which was beneficial to increasing electrical conductivity and reducing the charge of the metallic phase simultaneously. In addition, the proper dispersion of nanoparticles acts as an interconnected path for the flow of electrons during electrochemical evaluations. In addition, the exposed Ag atoms on the surface may significantly enhance the electrical conductivity of the bimetallic surface, improving electron transport, an essential step in electrocatalysis [47]. Both samples show a homogeneous distribution and adequate concentration of nanoparticles on the support material without aggregation, indicating the effectiveness of the synthesis method for obtaining dispersed bimetallic nanoparticles and a good sample preparation for further analysis.

Figure 4b,d shows the EDX-SEM analysis of AgPt (1:1) and AgPt (1:3) BNPs, respectively. The spectra showed the presence of Ag, Pt, and C from the samples, Cu and Al from the holder samples, and low detection of the Cl element from the remainder of the precursor salts used. From the EDX analysis, elementary information and atomic (at%) and weight (wt%) percentages were easily obtained. In addition, the analysis revealed that the AgPt BNPs maintained the elemental ratio of (1:1) and (1:3). These results agree with the ratios of the starting precursor salts and those obtained by XRD, UV-Vis, and STEM techniques.

Prior to analyzing the catalytic activity toward the ethanol oxidation reaction (EOR) on the as-prepared AgPt/C electrocatalysts, cyclic voltammetry (CV) curves were performed in 0.3 M KOH solution as an alkaline electrolyte in the absence of ethanol at a scan rate of 50 mVs^−1^, to gather information on the electrode surface behavior. The electrochemical profiles were recorded over a potential window of −0.80 to 0.85 V vs. NHE to include the processes of oxidation/reduction of both metal components.

Figure 5a displays the CV curves for AgPt (1:1)/C and AgPt (1:3)/C electrocatalysts, together with their monometallic counterparts (Ag/C and Pt/C). The Pt/C electrocatalyst showed a well-known electrochemical profile. The hydrogen desorption (H_des_) peaks were at a lower potential. Beyond the double-layer capacitive current, the Pt-oxide region was observed at around −0.05 V vs. NHE; these anodic processes occurred in the forward scan. The expected reduction of surface oxide from −0.05 to −0.50 V vs. NHE and the adsorption of hydrogen (H_ads_) at about −0.68 V vs. NHE occurred in the backward scan. For the Ag/C electrocatalyst, H_ads_/H_des_ peaks were not observed due to its electron configuration. This element had a weak hydrogen chemisorption ability, the same as Au and Cu (group 11 of the periodic table). However, when the forward potential scan reached 0.15 V vs. NHE, the first stage of formation of silver oxides started [48,49,50]. Two oxidation regions (Ox_1_ and Ox_2_) were observed. The Ox_1_ zone between 0.15 V vs. NHE and 0.60 V vs. NHE was due to Ag (I) oxide formation, a multi-stage process; thus, more than one signal was noticed. According to the present literature, Ag_2_O (or AgOH) monolayers originate first and then, through nucleation and a 3D growth mechanism, Ag_2_O multilayers. The Ox_2_ region was due to the subsequent oxidation of Ag_2_O to AgO and the direct oxidation of Ag to Ag (II) oxide through a two-electron process, which co-occurred simultaneously in the same potential range between 0.70 V vs. NHE and 0.85 V vs. NHE. Surprisingly, a small anodic peak (*Ox_3_) was observed at 0.66 V vs. NHE when reversing the potential scan direction. It has been associated with an autocatalytic process in which Ag (I) oxide is converted to Ag (II) oxide [48]. This means that the surface forms AgO multilayers through a non-stoichiometric oxidation process and is a mixed oxide. In addition, the reduction of the silver oxides takes place. The two cathodic peaks (R_2_ and R_1_) were conjugated with the oxidation signals, which were verified to progressively increase the anodic reverse potential limit. Figure 5b shows the relationship between the anodic and cathodic peaks of AgPt (1:1)/C. Thus, for all Ag-based samples, the cathodic peaks corresponded to the reduction of AgO to Ag_2_O (R_2_) and Ag_2_O to Ag (R_1_).

All oxidation/reduction processes of pure Pt and Ag were present in the bimetallic electrocatalysts. In essence, these electrochemical reactions occurred at or near the catalyst surface, suggesting that both metals are electrochemically accessible regardless of the elemental composition. They were dispersed on the surface of the BNPs, supporting the presence of an alloy structure. In addition, from Figure 5a, a negative shift in the Pt oxide reduction peak may be observed on both AgPt surfaces; consequently, the bimetallic surface transforms more oxygen species because they require less energy than pure Pt/C. As a complement to the analysis, Figure 5b displays an increase in the current density of the Pt oxide reduction peak. The hydrogen underpotential deposition (H_UPD_) region did not change as the reverse potential limit increased. This behavior supports the synergy between the Ag and Pt elements. Thus, the generation and reduction of more Pt oxides were favored when the oxidation of Ag was achieved.

The electrochemical active surface area (ECSA) was determined using CV curves similar to a powerful surface-sensitive technique. For pure Pt and AgPt bimetallic surface, the ECSA was determined on the charge associated with the Pt-H_UPD_ region, which is based on the adsorption/desorption of H atoms. Meanwhile, for pure Ag, the ECSA was determined by the coulomb charge of the redox reaction of the surface metal. This is the interaction between the surface atoms and oxygenated species, so the reduction peak of one monolayer of metal oxides was integrated. The resulting ECSA values are shown in Table 1. AgPt (1:3)/C presented an ECSA of 19.43 m^2^ g^−1^, which was higher than AgPt (1:1)/C (15.50 m^2^ g^−1^). Although the AgPt (1:3) BNPs had a larger size, the lower ECSA to AgPt (1:1)/C was attributed to introducing more Ag atoms on the surface, which was inactive toward the H_UPD_ and may have partially blocked the Pt sites. This represents a challenge for the surface modification strategy, tailoring the local electronic structure to boost the EOR catalytic performance but not represent a significative sacrifice of the ECSA [51].

The catalytic response in the presence of ethanol was explored by CV curves performed in 1.0 M ethanol solution in alkaline media (0.3 M KOH) at a scan rate of 20 mVs^−1^. The Ag surface failed to interact actively with ethanol molecules. Since it was catalytically inactive for ethanol oxidation, the current density was normalized by the Pt charge to obtain the mass activity (MA), which mainly reflects the cost efficiency between similar electrocatalysts, considering that all atoms of the particle were active sites. Figure 6a displays the catalytic behavior toward the EOR of AgPt (1:1)/C and AgPt (1:3)/C electrocatalysts, along with commercial Pt/C as a reference. The current density of the forward peak indicates the maximum reaction rate; thus, it was a principal parameter for insight into catalytic properties. The AgPt (1:3)/C presented an enhanced MA of 0.62 A mg^−1^ and for AgPt (1:1)/C of 0.29 A mg^−1^, while the commercial Pt/C was 0.16 A mg^−1^. This result indicates an MA for AgPt (1:3)/C of 3.9-fold higher than the commercial Pt/C results. This confirms that alloying with a lower-cost metal in an adequate composition is an effective approach to achieve superior mass activity and thus cost-effective bimetallic electrocatalysts, using an engineering perspective to develop electrochemical devices. However, to evaluate the intrinsic electrochemical performance of the catalyst surface, the specific activity (SA) was obtained, and the corresponding CV curves are shown in Figure 6b. The AgPt (1:3)/C presented an enhanced SA of 3.19 mA cm^−2^ and for AgPt (1:1)/C of 1.86 mA cm^−2^, while the commercial Pt/C was 1.15 mA cm^−2^. This result indicates a SA for AgPt (1:3)/C of 2.8-fold higher than commercial Pt/C. The MA and SA are also present in Figure 6c for better visualization and are reported in Table 1. Likewise, we noticed new properties, such as the AgPt (1:3)/C electrocatalyst presenting an onset potential with more negative values than the rest of the materials. The dashed line in Figure 6d indicates that AgPt (1:3)/C reaches a SA of 0.1 mA cm^−2^, 160 mV earlier than the commercial Pt/C, indicating a lower energy barrier of the reaction, so the EOR was more facile on the AgPt (1:3)/C surface.

A review of the literature supports the advancement of AgPt (1:3)/C compared to other transition metals as a co-catalyst element toward the EOR. For example, the forward peak current density of the octahedral Pt_2.3_Ni/C with an average size of 10 nm displayed an SA of 1.46 mA cm^−2^, and the authors explained that the improved catalytic performance implies the octahedral shape and alloying effect [52]. In other work, PtAg porous nanotubes with delicate 3D porous wall structures showed a specific current density of 1.97 mA cm^−2^ [45]. The previous materials showed lower performance than the AgPt (1:3)/C prepared in this work. With a similar current density, the ethanol oxidation performance of the ultrathin PtRu NWs/C was 3.78 mA cm^−2^ [53]. Additionally, the working group evaluated the PdPt/C electrocatalyst, showing an SA of 3.27 mA cm^−2^ [54]; however, Ru and Pd are expensive noble metals. Therefore, the AgPt (1:3)/C electrocatalyst has competitive performance toward the OER with the advantage of being a cost-effective material. More information about comparing the catalytic activity of ethanol oxidation using Pt-based materials is shown in Appendix A.

As is customary, the stability of the electrocatalysts was investigated using chronoamperometry (CA) measurements under alkaline conditions. Figure 7a indicates the highest catalytic activity maintained on AgPt (1:3)/C after anodic polarization at −0.25 V (vs. Hg/HgO) for 3600 s. One point of interest was at 360 s (6.5 min). Both bimetallic electrocatalysts exhibited the same EOR oxidation current density, showing 37% retention of catalytic activity compared to 24% for commercial Pt/C. However, 14.8%, 9.6%, and 2.5% of the initial performance was sustained on AgPt (1:3)/C, AgPt (1:1)/C, and commercial Pt/C at the end of the CA test, respectively. During the electrocatalytic process, intermediate species, such as CO-like species, were generated, most likely due to the oxidation mechanism on Pt surfaces [45]. Adsorption of CO on the active site of the catalytic surface generates CO coverage that easily poisons the electrocatalyst. CO-like binds strongly to the active surface, causing a decrease in catalytic stability. More and more active sites are blocked as the reaction progresses if the electrocatalyst cannot perform the oxidative desorption of the CO-like species in a timely manner, resulting in a rate of decrease in current density. Therefore, presenting a higher current density implies a high number of active sites (higher ECSA) and a balance of binding energies between ethanol molecules and oxygenated adsorbed species (OH_ads_ and CO_ads_) to observe reasonable EOR activity. Consequently, the OH_ads_ species can proceed from the H_2_O adsorbed on the active site, which gives rise to the formation of OH^-^ species. According to the volcano plot, Ag binds weakly to oxygen species. Meanwhile, the strength of Pt is strong [55]. Combining both metals in a suitable elemental arrangement could tune the O-binding for the EOR through an ensemble effect, where specific group arrangements of atoms on the surface are needed to act as active sites, so that both metals interact in synergy, promoting faster kinetics.

To verify the tolerance to poisonous species, following the CA test, with enough time to observe a stable state in the electrocatalytic response, a couple of voltammetry cycles were performed in a fresh alkaline solution for the same potential window and from a lower potential limit toward the anodic direction are displayed in Appendix A. It is assumed that the visualization of additional peaks or regions is due to the presence of active species that remained adsorbed on the electrode throughout the stability test. In this case, the effect of carbonaceous intermediates may be reflected in surface poisoning and correlated with the resulting stability test. Thus, the superior stability of AgPt (1:3)/C can be seen in Figure 7b, where a close-up of the anodic scan of the CV curves of Appendix A is displayed to analyze the oxidative desorption of carbonous species on the different electrocatalysts. The main peak potential on AgPt (1:3)/C and AgPt (1:1)/C at −0.21 V vs. NHE was more negative than the value for Pt/C (−0.16 V vs. NHE); this indicates that the removal of CO-like species is more facile on the bimetallic surface, which is caused by the weaker CO adsorption strength due to the possible change in the electronic structure by the incorporation of Ag atoms into the Pt structure. However, the cover layer was higher on AgPt (1:1)/C. Therefore, the highest residual EOR activity on AgPt (1:3)/C may have arisen from the lower accumulation of intermediate carbonaceous species, such as CO, due to its higher ECSA and more effective oxidation step for such species. In addition, a prominent and broad CO-like oxidation region on Pt/C might correspond to the two types of CO_ads_ species due to its binding energy, which is stronger than AgPt/C samples. In the second cycle, the CO oxidation region is not observed; thus, a standard CV curve is observed for all materials (Appendix A).

As a strategy, this analysis supports the introduction of a smaller amount of foreign metal, such as Ag, on the active surface. Ag has a beneficial effect on the catalytic performance of the system due to the electrochemical behavior of these alloys, depending on their synthesis conditions. A successful modification of the surface composition can tune the electrocatalytic properties of the electrocatalyst by creating a high number of active sites (high ECSA) and optimizing the adsorption of intermediates. In this case, AgPt (1:3)/C present the highest Pt content; this ensures sufficient active surface sites, and the small portion of Ag does not sacrifice the ECSA. Instead, it shows a local change in the electronic structure that boosts the EOR through weakened Pt-intermediate bonding strength, such as Pt-CO, to ease the removal of the catalytic surface. Therefore, Ag plays an essential role in adjusting the inherent adsorption energies of the new catalytic surface.

Long-term stability testing is crucial for using electrocatalysts in fuel cells. In this sense, accelerated electrochemical degradation tests were performed on AgPt (1:3)/C to show the best catalytic performance and commercial Pt/C as a control electrocatalyst. The electrocatalysts were evaluated during 2500 potential scan cycles at a scan of 100 mVs^−1^ toward the EOR in alkaline media. Figure 8a shows the retention of the catalytic activity for the evaluated cycles (initial, 250, 500, 1000, and 2500), obtained from the forward peak of the CV curves recorded at 20 mVs^−1^ for ethanol oxidation. AgPt (1:3)/C displays a lower degradation rate than Pt/C. Thus, the catalytic activity at the end of the test of 2.49 mA cm^−2^ for AgPt (1:3)/C and 0.63 mA cm^−2^ for commercial Pt/C (Figure 8b) corresponds to a loss of catalytic activity over initial values of 22% and 45%, respectively. Therefore, higher cyclic stability was confirmed for AgPt (1:3)/C, which suggests that the combination of two elements that are complementary due to their H- and O-binding energy. Ag is weak compared to Pt, which is strong, originates a new surface, and through a synergic effect, can express a balance in the adsorption energies, accomplishing the design of electrocatalysts with high catalytic activity and durability, simultaneously.

## 4. Conclusions

In the present work, a chemical synthesis methodology based on the polyol method was successfully developed to obtain AgPt BNPs in different atomic compositions (1:1 and 1:3). The XRD technique determined that the AgPt BNPs present an fcc crystalline structure in an alloy-type configuration. Through the UV-Vis technique, the changes in the optoelectronic properties of the nanoalloys produced by the incorporation of Ag and Pt atoms into the Ag growth seeds were determined. The extinction of the SPR band characteristic of Ag seeds was defined with the increase in the atomic percentage of Pt, confirming the formation of AgPt nanoalloys. SEM and HAADF-STEM determined the particle size, morphology, and distribution of the nanoalloys, obtaining an average particle size of 6 and 10 nm for the ratios 1:1 and 1:3, respectively. The nanoparticles in both systems presented a good distribution along the support material (Vulcan XC-72 carbon). On the other hand, the particles exhibited a truncated octahedral morphology with exposed {100} and {111} surface planes related to highly stable surface energy densities and selectivity for specific catalytic reactions. The synergistic interaction between Ag and Pt was studied as an electrocatalyst for ethanol oxidation, verifying that the ratio between Pt and Ag significantly affected the electrocatalytic performance, although pure Ag has no activity toward the EOR. AgPt (1:3)/C presented a new catalytic surface exhibiting a MA/SA 3.9/2.8 times higher than the commercial Pt/C, as well as superior CO-like tolerance and the best stability. Alloying Pt with Ag is an attractive strategy for designing cost-effective electrocatalysts. Therefore, AgPt (1:3)/C has the competitive potential of being used as an electrode material for fuel cell applications.

## Figures and Tables

**Figure 1 nanomaterials-13-01396-f001:**
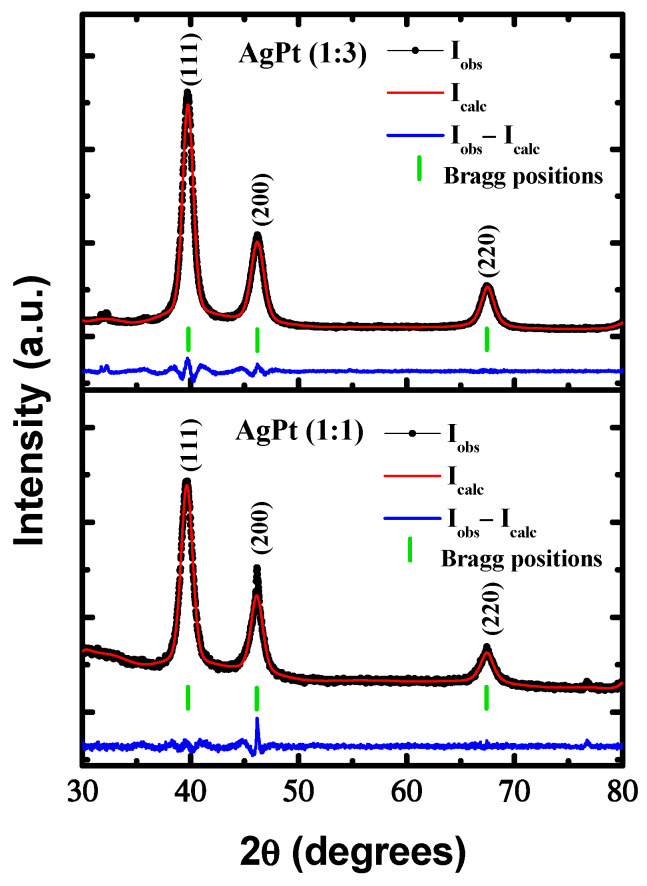
XRD patterns obtained from AgPt bimetallic nanoparticles with different atomic ratios.

**Figure 2 nanomaterials-13-01396-f002:**
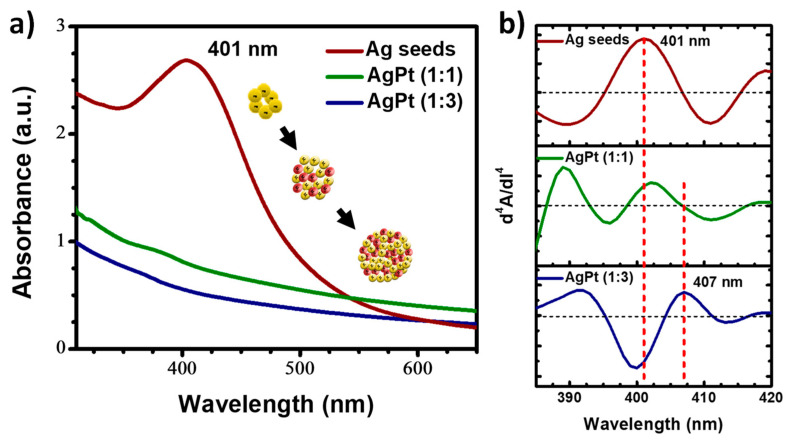
(**a**) UV-Vis absorption spectra and (**b**) fourth-derivative absorption spectra of the obtained Ag seeds, AgPt (1:1), and AgPt (1:3) BNPs.

**Figure 3 nanomaterials-13-01396-f003:**
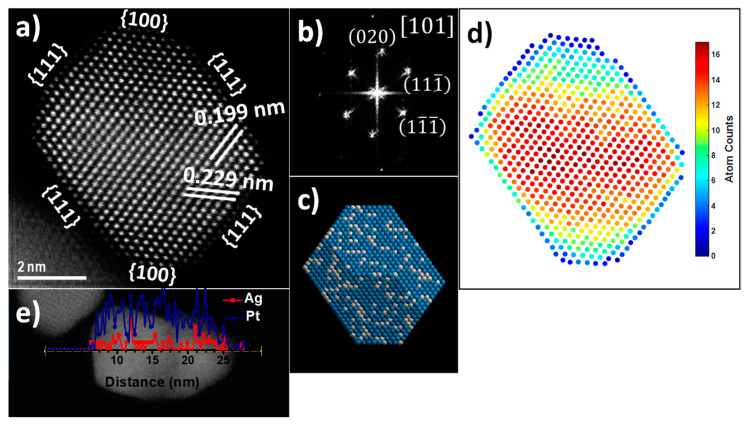
(**a**) HAADF-STEM image of AgPt (1:3) BNP along the [101] zone axis, (**b**) FFT pattern, (**c**) truncated octahedron model along the same direction, (**d**) atom counts, and (**e**) Ag and Pt elemental profiles along the line across the BNPs.

**Figure 4 nanomaterials-13-01396-f004:**
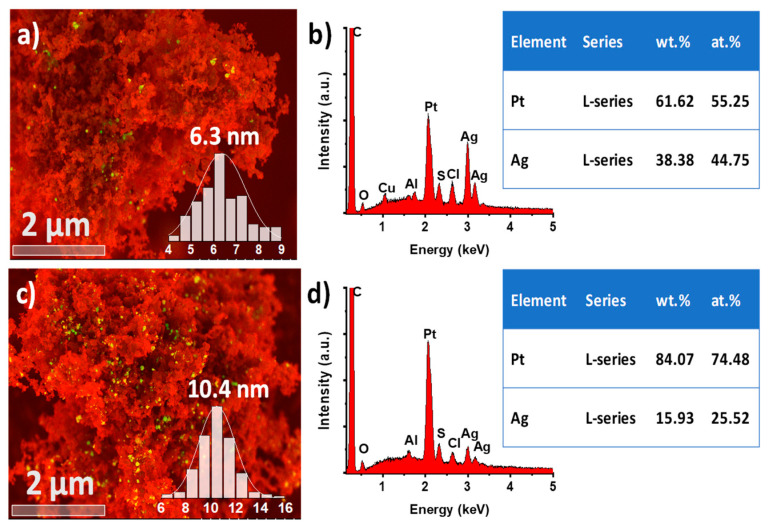
SEM image with its corresponding EDX analysis of (**a**) and (**b**) AgPt (1:1), and (**c**) and (**d**) AgPt (1:3) BNPs.

**Figure 5 nanomaterials-13-01396-f005:**
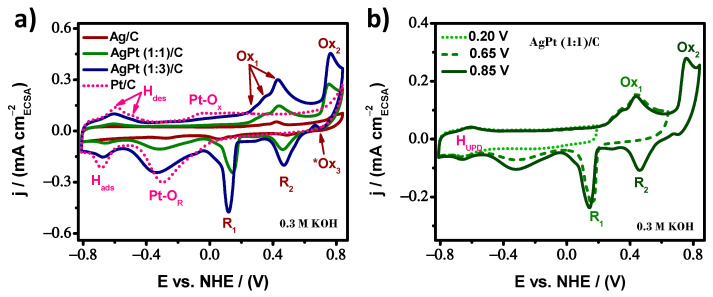
Cyclic voltammograms profiles in 0.3 M KOH solution at 50 mVs^−1^; (**a**) for AgPt/C system, and (**b**) at the different anodic potential limits for AgPt (1:1)/C.

**Figure 6 nanomaterials-13-01396-f006:**
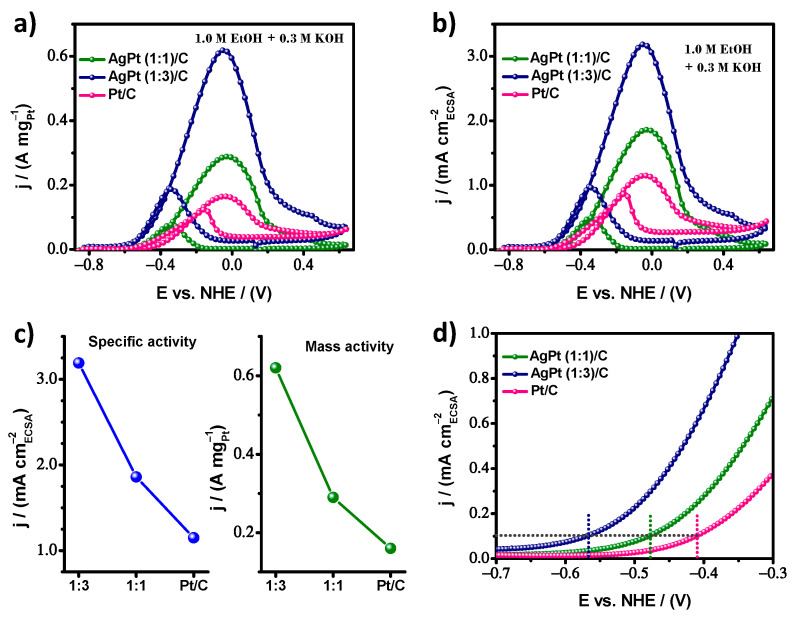
Cyclic voltammograms in 0.3 M KOH + 1.0 M ethanol solution at 20 mV s^−1^ normalized for (**a**) mass activity (MA), (**b**) specific activity (SA), (**c**) mass activity (MA) and specific activity (SA) at the maximum point of the reaction rate, and (**d**) close-up of the initial region of the ethanol oxidation for AgPt (1:1)/C, AgPt (1:3)/C, and Pt/C electrocatalysts.

**Figure 7 nanomaterials-13-01396-f007:**
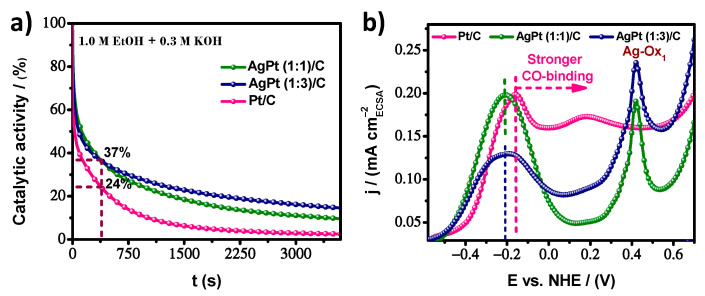
(**a**) Residual catalytic activity based on chronoamperometry (CA) measurement at −0.25 V vs. Hg/HgO for ethanol oxidation reaction in alkaline media and (**b**) close-up on cyclic voltammograms curves in the anodic scan for AgPt (1:1)/C, AgPt (1:3)/C, and commercial Pt/C electrocatalysts.

**Figure 8 nanomaterials-13-01396-f008:**
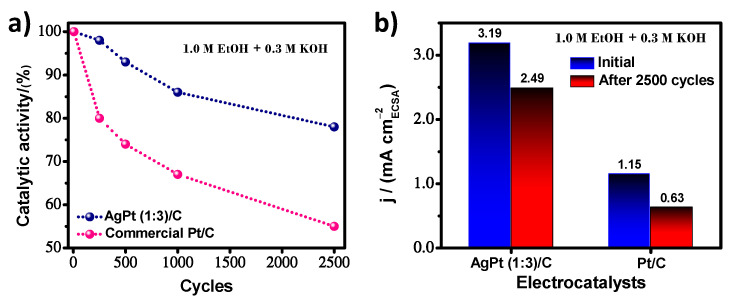
(**a**) Evaluation of the percentage of catalytic activity as a function of the number of cycles during the accelerated electrochemical degradation test, and (**b**) comparison of the initial and final catalytic activity for AgPt (1:3)/C and commercial Pt/C electrocatalysts.

**Table 1 nanomaterials-13-01396-t001:** Electrochemically active surface area, mass activity, and specific activity of the different electrocatalysts.

Electrocatalyst	ECSA m^2^/g	MA A/mg	SA mA/cm^2^
AgPt (1:1)/C	15.4 ± 0.3	0.29	1.86
AgPt (1:3)/C	19.5 ± 0.3	0.62	3.19
Pt/C	14.5 ± 0.2	0.16	1.15
Ag/C	11.1 ± 0.1	-	-

## Data Availability

Not applicable.

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
