# Peer review of "Bimetallic AgPt Nanoalloys as an Electrocatalyst for Ethanol Oxidation Reaction: Synthesis, Structural Analysis, and Electro-Catalytic Activity"

_nanomaterials, 2023, doi:10.3390/nano13081396_

Round 1
Reviewer 1 Report
In this work, the authors synthesized bimetallic AgPt nanoalloys as an electrocatalyst for ethanol oxidation reaction, and performed structural analysis and electrocatalytic activity. They prepared 2 different samples of Ag/Pt alloys, which have different atomic ratio. characterization method and results are sound to draw a proper conclusion and they performed systematic electrochemical reaction to reveal the role for each of Ag and Pt. Therefore, the reviewer think this manuscript is suitable for the publication in Nanomaterials journal, after addressing some minor comments.
1. Author evaluate the size of NPs through SEM images but if the size is less than 10 nm, it is recommended to analyze and provide some TEM images.
2. Quantitative elemental analysis via EDX might be inaccurate. the reviewer recommends to confirns the amount of each element (Ag and Pt) via ICP analysis.
Author Response
In this work, the authors synthesized bimetallic AgPt nanoalloys as an electrocatalyst for ethanol oxidation reaction, and performed structural analysis and electrocatalytic activity. They prepared 2 different samples of Ag/Pt alloys, which have different atomic ratio. characterization method and results are sound to draw a proper conclusion and they performed systematic electrochemical reaction to reveal the role for each of Ag and Pt. Therefore, the reviewer think this manuscript is suitable for the publication in Nanomaterials journal, after addressing some minor comments.
- Author evaluate the size of NPs through SEM images but if the size is less than 10 nm, it is recommended to analyze and provide some TEM images.
For a better analysis of the size and morphology of the AgPt nanoalloys, we included STEM micrographs in the supporting information.
- Quantitative elemental analysis via EDX might be inaccurate. the reviewer recommends to confirns the amount of each element (Ag and Pt) via ICP analysis.
The authors agree that an ICP analysis of AgPt nanoalloys is the best technique to confirm each element's amount. However, this analysis is out of our possibilities. On the other hand, we perform different EDX analyses along the samples to reduce the error rate on the quantitative elemental analysis.
AgPt (1:3)
|
Element |
at%_1 |
at%_2 |
at%_3 |
at%_4 |
at%_avg. |
|
Ag |
23.80 |
26.53 |
26.81 |
24.93 |
25.52 |
|
Pt |
76.20 |
73.47 |
73.47 |
75.07 |
74.48 |
AgPt (1:1)
|
Element |
at%_1 |
at%_2 |
at%_3 |
at%_4 |
at%_avg. |
|
Ag |
41.90 |
53.33 |
33.49 |
50.28 |
44.75 |
|
Pt |
58.10 |
46.67 |
66.51 |
49.72 |
55.25 |
Reviewer 2 Report
In this paper, AgPt nanoalloys was synthesized by a chemical polyol method. Atomic compositions of the AgPt nanoalloys can be tuned by adjusting the molar ratios of the precursors. Electrochemical tests show the as prepared AgPt nanoalloys hold potential as electrocatalyst for ethanol oxidation. The result and discussion are reasonable and acceptable. However, there are still some places which should be carefully issued.
Comments:
1. What is the novelty of this manuscript? Because similar method (synthesis of Au@Pt) has been reported in Nanomaterials 2019, 9, 1644. 56.
2. The author should explain why choose the ratio of Ag and Pt elements as 1:1 and 1:3?
3. The electrocatalytic performances of AgPt nanoalloys should be compared with other previously reported Pt based electrocatalyst, such as the previous Au@Pt electrocatalyst.
4. TEM results with low magnification need to be given to show the morphology of AgPt nanoalloys.
Author Response
Comments:
- What is the novelty of this manuscript? Because similar method (synthesis of Au@Pt) has been reported in Nanomaterials 2019, 9, 1644. 56.
Au@Pt and AgPt represent two different approaches; the co-catalyst element is dissimilar (Au or Ag). Likewise, the first system was based on two noble metals and a core-shell structure. The second one was about an alloy structure with a low-cost element that can enter the field of cost-effective electrocatalysts. Therefore, the different structures and elemental arrangement will give each system unique synergistic effects. In addition, the alcohol used as fuel was different (methanol vs. ethanol).
On the other hand, a prerequisite in exploring advanced electrocatalysts is to synthesize nanoparticles with uniform characteristics (size, morphology) since the study of their catalytic properties is favored. Therefore, having synthesis methods that achieve good control of the features of the nanoparticles is essential. So, the polyol method, as mentioned in the article and verified by the working group, is an easy and flexible synthesis route that can be successfully adapted to design different bimetallic systems.
- The author should explain why choose the ratio of Ag and Pt elements as 1:1 and 1:3?
A general scan but over a wide range of composition for AgPt could be AgXPt100-X (where X= 100, 75, 50, 25), which leaves the synthesis of Ag, AgPt (3:1), AgPt (1:1), and AgPt (1:3). However, AgPt (3:1) BNPs were discarded since it is known that Ag does not show activity for the oxidation of ethanol. The design of an alloy-type structure, where both metals are electrochemically accessible, would lead to a loss of active sites. Remember that advanced electrocatalysts seek a high number of active sites and boost the intrinsic activity of the active phase. Therefore, introducing a larger number of Ag atoms is likely not beneficial to obtain a significant performance towards the EOR. On the other hand, a greater amount of Pt could no longer be profitable. This is supported by theoretical studies that determined the optimal elemental composition of Pt3Ag (REF https://doi.org/10.1016/j.electacta.2018.04.102).
- The electrocatalytic performances of AgPt nanoalloys should be compared with other previously reported Pt based electrocatalyst, such as the previous Au@Pt electrocatalyst.
In the updated version of this article, a table comparing other Pt-based electrocatalysts for ethanol oxidation is now included (Supporting Information). However, the previously reported Au@Pt electrocatalyst was not included because it was evaluated for methanol oxidation. However, AgPt is compared to another electrocatalyst previously developed by the working group.
- TEM results with low magnification need to be given to show the morphology of AgPt nanoalloys.
The new version includes low-magnification micrographs of AgPt nanoalloys showing their morphology in the supplementary document.
Reviewer 3 Report
The work "Bimetallic AgPt nanoalloys as an electrocatalyst for ethanol oxidation reaction: synthesis, structural analysis and electrocatalytic activity" provides detailed information on promising catalysts for the ethanol oxidation reaction in an alkaline medium. The work is devoted to the topical issue of improving catalysts for the green energy development. Nevertheless, a number of questions and comments on the work should be noted:
Why is the composition of PtAg alloys for the obtained samples not determined by Vegard's law? It is possible that silver is not completely included in the composition of bimetallic NPs.
For a more complete study of bimetallic nanoparticles, it is useful to use the XPS method, which will allow one to determine the valence state of atoms on the surface.
What is the error in determining the value of ESA? ESA values such as 19.43 m2 g-1 cannot be determined with such high precision and should be rounded to reasonable values.
What Pt/C catalyst is used for comparison? Why are its structural characteristics not specified, for example, the size of platinum nanoparticles? Why does it have such a low area? Typically, commercial materials with high ESA values are used.
AgPt/C catalysts in an alkaline medium are well known, what is new proposed by the authors? Why are the authors studying these particular compositions for the Pt-Ag system?
Author Response
The work "Bimetallic AgPt nanoalloys as an electrocatalyst for ethanol oxidation reaction: synthesis, structural analysis and electrocatalytic activity" provides detailed information on promising catalysts for the ethanol oxidation reaction in an alkaline medium. The work is devoted to the topical issue of improving catalysts for the green energy development. Nevertheless, a number of questions and comments on the work should be noted:
Why is the composition of PtAg alloys for the obtained samples not determined by Vegard's law? It is possible that silver is not completely included in the composition of bimetallic NPs.
The new version discusses the composition of AgPt alloys determined by Vegard's law.
For a more complete study of bimetallic nanoparticles, it is useful to use the XPS method, which will allow one to determine the valence state of atoms on the surface.
We fully agree that XPS is a helpful technique to determine the valence state of atoms on the surface and can contribute valuable information to the investigation. However, unfortunately, it is out of our hands to do this technique because the equipment we have within reach to carry out this technique is out of service.
What is the error in determining the value of ESA? ESA values such as 19.43 m2 g-1 cannot be determined with such high precision and should be rounded to reasonable values.
Four CV curves were recorded with a different catalytic ink in order to verify the reproducibility of the electrocatalysts behavior and estimate of the experimental error of the ECSA values. The currently results were modified in the new version of this article.
|
Electrocatalyst |
ECSA-1 |
ECSA-2 |
ECSA-3 |
ECSA-4 |
Average |
Error |
|
|
m2/g |
|
||||
|
AgPt (1:1)/C |
15.5 |
15.0 |
15.6 |
15.4 |
15.4 |
0.3 |
|
AgPt (1:3)/C |
19.4 |
19.2 |
19.9 |
19.6 |
19.5 |
0.3 |
|
Pt/C |
14.3 |
14.7 |
14.3 |
14.5 |
14.5 |
0.2 |
|
Ag/C |
11.0 |
11.1 |
11.3 |
11.0 |
11.1 |
0.1 |
Consequently, the SA values suffer a slight modification that was also verified by means of the ECSA average.
What Pt/C catalyst is used for comparison? Why are its structural characteristics not specified, for example, the size of platinum nanoparticles? Why does it have such a low area? Typically, commercial materials with high ESA values are used.
The new version now includes information on the Pt/C catalyst, and the product reference is in the materials and method section. In addition, an SEM-SE low-magnification micrograph showing the particle distribution is now included in the supplementary document.
The commercial Pt was purchased from Sigma-Aldrich (Sigma-Aldrich, St. Louis, MO, USA). It is platinum on graphitized carbon (20% Pt/Vulcan). The SEM image shows Pt nanoparticles with an average size of 5 nm, but some regions with several aggregations can be observed, which reduces the ECSA of the active phase. The ECSA value in this work is in agreement with other reported, such as, the resulted ECSA of the commercial Pt black (11.34 m2 g-1) (ref https://doi.org/10.1016/j.ijhydene.2015.11.021).
AgPt/C catalysts in an alkaline medium are well known, what is new proposed by the authors? Why are the authors studying these particular compositions for the Pt-Ag system?
AgPt/C catalysts in an alkaline medium are well known, what is new proposed by the authors?
The present research seeks viable alternatives for electrocatalysts that meet the cost-effectiveness premise without compromising durability, selectivity, stability, and high ECSA values towards EOR. The design of Pt-based structures in an alloy arrangement with lower-cost elements such as Ag can improve the accessibility of the materials used. However, it is crucial to study the behavior based on the composition and distribution of the elements that make up the alloy, looking for the correct balance and operation between components. According to the present study, it was shown that Ag by itself does not show activity for the oxidation of ethanol. Therefore, introducing a large number of Ag atoms is probably not beneficial to obtaining a significant yield towards the EOR. However, through the reported electrochemical studies, it was proven that forming AgPt nanoalloys in low Ag percentages can improve the stability of the electrocatalyst towards poisoning by intermediate species, promoting a high number of active sites and enhancing the intrinsic activity of the phase active of the platinum.
Why are the authors studying these particular compositions for the Pt-Ag system?
A general scan but over a wide range of composition for AgPt could be AgXPt100-X (where X= 100, 75, 50, 25), which leaves the synthesis of Ag, AgPt (3:1), AgPt (1:1), and AgPt (1:3). However, AgPt (3:1) BNPs were discarded since it is known that Ag does not show activity for the oxidation of ethanol. The design of an alloy-type structure, where both metals are electrochemically accessible, would lead to a loss of active sites. Remember that advanced electrocatalysts seek a high number of active sites and boost the intrinsic activity of the active phase. Therefore, introducing a larger number of Ag atoms is likely not beneficial to obtain a significant performance towards the EOR. On the other hand, a greater amount of Pt could no longer be profitable. This is supported by theoretical studies that determined the optimal elemental composition of Pt3Ag (REF https://doi.org/10.1016/j.electacta.2018.04.102).
Round 2
Reviewer 2 Report
All the questions have been issued. I suggest it could be accepted for publication.
Reviewer 3 Report
All comments have been corrected and the article can be accepted for publication.